# Description of Antimicrobial-Resistant *Escherichia coli* and Their Dissemination Mechanisms on Dairy Farms

**DOI:** 10.3390/vetsci10040242

**Published:** 2023-03-23

**Authors:** Jonathan Massé, Ghyslaine Vanier, John M. Fairbrother, Maud de Lagarde, Julie Arsenault, David Francoz, Simon Dufour, Marie Archambault

**Affiliations:** 1Regroupement FRQNT Op+lait, Saint-Hyacinthe, QC J2S 2M2, Canada; 2Groupe de Recherche sur les Maladies Infectieuses en Production Animale (GREMIP), Saint-Hyacinthe, QC J2S 2M2, Canada; 3Department of Pathology and Microbiology, Faculty of Veterinary Medicine, Université de Montréal, Saint-Hyacinthe, QC J2S 2M2, Canada; 4WOAH Reference Laboratory for *Escherichia coli*, Faculty of Veterinary Medicine, Université de Montréal, Saint-Hyacinthe, QC J2S 2M2, Canada; 5Department of Clinical Sciences, Faculty of Veterinary Medicine, Université de Montréal, Saint-Hyacinthe, QC J2S 2M2, Canada

**Keywords:** antimicrobial resistance, whole genome sequencing, dairy cattle, fluoroquinolone, third-generation cephalosporin, multidrug resistance, horizontal transfer, plasmid, clone, mobile genetic element

## Abstract

**Simple Summary:**

There is little information about antimicrobial resistance and the transmission of this resistance in dairy cattle. The aim of this work is to use cutting-edge technology (whole genome sequencing) to characterize antimicrobial resistance of bacteria (*Escherichia coli*) retrieved from the manure of dairy cattle and to determine how this resistance spreads among the *E. coli* population on dairy farms in Québec, Canada. It detects some resistance genes for antimicrobials considered to be of high priority and critical importance to human medicine. Some of these genes are situated close to each other as well as in the vicinity of some elements for transfer between bacteria. This suggests that bacteria can share these critical antimicrobial resistance genes on dairy farms. Furthermore, the same bacterium was found on farms located up to more than 100 km apart, suggesting transmission across dairy farms. An understanding of this dissemination mechanism will allow for the elaboration of fighting strategies against these resistant bacteria.

**Abstract:**

Despite its importance in veterinary medicine, there is little information about antimicrobial resistance (AMR) and its transmission in dairy cattle. The aim of this work is to compare AMR phenotypes and genotypes in resistant *Escherichia coli* and to determine how the resistance genes spread among the *E. coli* population on dairy farms in Québec, Canada. From an existing culture collection of *E. coli* isolated from dairy manure, a convenient selection of the most resistant isolates (a high level of multidrug resistance or resistance to broad-spectrum β-lactams or fluoroquinolones) was analyzed (*n* = 118). An AMR phenotype profile was obtained for each isolate. Whole genome sequencing was used to determine the presence of resistance genes, point mutations, and mobile genetic elements. In addition, a subset of isolates from 86 farms was taken to investigate the phylogenetic relationship and geographic distribution of the isolates. The average agreement between AMR phenotypes and genotypes was 95%. A third-generation cephalosporin resistance gene (*bla_CTX-M-15_*), a resistance gene conferring reduced susceptibility to fluoroquinolones (*qnrS*1), and an insertion sequence (ISKpn19) were detected in the vicinity of each other on the genome. These genes were harbored in one triplet of clonal isolates from three farms located >100 km apart. Our study reveals the dissemination of resistant *E. coli* clones between dairy farms. Furthermore, these clones are resistant to broad-spectrum β-lactam and fluoroquinolone antimicrobials.

## 1. Introduction

The World Health Organization (WHO) has proclaimed antimicrobial resistance (AMR) to be one of the greatest current threats to global health [1]. It is generally accepted that AMR is directly associated with the use of antimicrobials [1]. AMR bacteria pose serious problems associated with treatment failures or the transmission of resistance. Among these, extended-spectrum β-lactamase (ESBL)-producing *Escherichia coli* are resistant to most third- and fourth-generation cephalosporins, which are classified as the highest priority critically important antimicrobials by the WHO [2]. Increasing detection of ESBL-producing *E. coli* strains in livestock has been noted [3,4,5,6], making them of particular concern because of the potential for the transfer of resistance to humans, especially those with direct exposure to food-producing animals [7]. Furthermore, clonal transmission of bacteria resistant to third-generation cephalosporins has been described in farm animals such as pigs [8] and chickens [9]. In addition, fluoroquinolones are an important class of broad-spectrum antimicrobials against many Gram-negative aerobes and are also classified as the highest priority critically important antimicrobials by the WHO. The prevalence of *E. coli* with reduced susceptibility toward fluoroquinolones remains low for livestock in North America [10,11]. However, quinolone-resistant *E. coli* are particularly problematic for human medicine [12] and clonal dissemination has been described [13]. Monitoring of AMR in bacteria relies on classic phenotypic tests described in standardized reference guides such as CLSI [14] or EUCAST [15]. In the last few years, whole genome sequencing (WGS) of bacteria has enriched the classic phenotypic techniques. A strong correlation between phenotypic and genotypic AMR has been reported for bacteria of human [16,17] and animal origin [18].

Our group has recently described antimicrobial usage on 101 dairy farms in Québec, Canada [19]. The AMR in *E. coli* from fecal samples was also investigated for these same farms [20]. The highest observed prevalence of AMR was towards tetracycline, sulfisoxazole, and streptomycin [20]. Preliminary WGS results in a limited number of isolates demonstrated the resistance genes associated with these resistance phenotypes [20]. However, the full potential of WGS was not exploited in this previous study. In addition to demonstrating the presence of resistance genes, WGS using short reads can, to a certain extent, evaluate the proximity of resistance genes [18] and the proximity of these genes to mobile genetic elements [21]. Furthermore, WGS can establish phylogenetic relatedness [22] and demonstrate clonal dissemination of resistant bacteria [23]. The present study characterizes the AMR in greater depth using a larger pool of multidrug-resistant, broad-spectrum β-lactam, or fluoroquinolone-resistant *E. coli* from dairy cattle manure. This will enhance our knowledge of bacterial resistance in dairy cattle.

The three hypotheses driving this study are: (1) multidrug-resistant or broad-spectrum β-lactam or fluoroquinolone-resistant *E. coli* isolated from dairy cattle manure possess resistance genes that encode phenotypic resistance; (2) these resistance genes are located in close proximity and in the vicinity of mobile genetic elements in the bacterial genome; and (3) there is clonal transmission of these resistant bacteria between dairy farms. The main aim of this work is to elucidate the genetic component of the resistance to broad-spectrum β-lactams and fluoroquinolones and to determine how these genes are spread in the *E. coli* population on dairy farms. 

## 2. Materials and Methods 

### 2.1. Culture Collection and Selection of Isolates

Isolates from this study originate from an observational cross-sectional study on 101 commercial dairy farms (Québec, Canada) which was previously described [20]. Selection of herds, fecal sample collection, and bacterial isolation are available elsewhere [20]. The culture collection included: 593 randomly selected indicator *E. coli* and 214 ESBL/AmpC-producing *E. coli* obtained with a selective protocol. The first two hypotheses (the presence of resistance genes and mobile genetic elements) were tested with a convenient selection of the most resistant isolates regrouped in collection A. The third hypothesis (clonal dissemination between farms) was tested with a standardized random selection of isolates regrouped in collection B. Collection A comprised 118 isolates selected as follows: (1) twenty-seven isolates from the indicator *E. coli* subset based on at least one of the three following criteria: resistant to seven or more antimicrobial classes (aminoglycosides and aminocyclitols were considered two different classes for this selection), resistant to ceftriaxone (MIC ≥ 2 μg/mL) or a high MIC to ciprofloxacin (MIC ≥ 0.064 μg/mL); (2) five isolates selected for an atypical phenotype from the ESBL/AmpC-producing *E. coli* subset: four isolates with an ESBL/AmpC phenotype and one isolate with the phenotype “other”, and (3) eighty-six isolates from collection B which comprised eighty-six isolates that were selected randomly (ALEA function in Excel), one per positive farm, from the ESBL/AmpC-producing *E. coli* subset. Fifteen farms did not demonstrate any ESBL/AmpC-producing *E. coli* and were excluded from further analyses.

### 2.2. Antimicrobial Resistance Phenotypes

The antimicrobial resistance phenotypes were previously described for isolates selected in the indicator *E. coli* subset (*n* = 27) [20]. The same protocol was used for the remaining 91 isolates selected in the ESBL/AmpC-producing *E. coli* subset. Briefly, the minimum inhibitory concentrations (MICs) for 20 antimicrobials were tested. As previously described [20], the isolates were defined as susceptible, intermediate, or resistant according to CLSI M100 [14] (human *Enterobacterales*), CLSI VET01S [24] (bovine respiratory pathogens), or CIPARS [11] breakpoints. The epidemiological cut-off value from the European Committee on Antimicrobial Susceptibility Testing (EUCAST) was used for neomycin (MIC ≥ 16 µg/mL was defined as resistant). There was no valid florfenicol breakpoint for *Enterobacterales* and the tested concentrations (0.25–8 µg/mL) did not include the European epidemiological cut-off of 16 µg/mL; therefore, no interpretation was attempted for this latter antimicrobial. For subsequent analyses, intermediate and resistant isolates were grouped together and labelled as resistant. For sensitivity and specificity, the phenotypic resistance defined by breakpoints was considered as the “gold standard” method. In addition to breakpoints described above, isolates were also classified by EUCAST epidemiological cut-off values. All values used for definition of AMR phenotypes are shown in the Appendix A. *Enterococcus faecalis* ATCC 29212, *Escherichia coli* ATCC 25922, *Staphylococcus aureus* ATCC 29213, and *Pseudomonas aeruginosa* ATCC 27853 were used as reference strains for batch controls. *Escherichia coli* ATCC 25922 was used as a daily control.

### 2.3. Whole Genome Sequencing

The QIAamp DNA Mini Kit for DNA was used to extract genomic DNA according to the manufacturer’s guidelines (Qiagen, Hilden, Germany). The Illumina Nextera XT DNA Library Preparation Kit was used, according to the manufacturer’s instructions. Whole genome sequencing was performed on the Illumina (San Diego, CA, USA) MiSeq platform with 2 × 300 paired-end runs. The assembled genomes were obtained using SPADES software. An assembly was rejected if the number of contigs was >400, if the N50 was <40,000, or if the number of contigs was between 300 and 400 and the N50 < 50,000. The quality of the genome assembly is presented in Appendix A. The sequences were submitted to NCBI and the bioproject ID is PRJNA716674. 

For collection A, the Center of Genomic Epidemiology (CGE) platform (http://www.genomicepidemiology.org/, accessed on 25 January 2023) was used for all the analyses. Resistance genes, point mutations, and replicons were searched using ResFinder 4.1 [25], PointFinder [26], and PlasmidFinder [27], respectively. An ID threshold ≥95% and the minimum length ≥85% were used for ResFinder. An ID threshold ≥ 95% and the minimum length ≥95% were used for PlasmidFinder. Mobile genetic elements and their association with antimicrobial resistance genes were searched using MGE finder [21]. Plasmid ST was searched using pMLST [27]. For some isolates, the resistance gene *bla_CMY_* was truncated in the FASTA files; in these cases, FASTQ files were used instead. 

For collection B, supplemental phylogenetic analyses were carried out. Phylogroups were determined with in silico PCR using the Clermont Typing platform (http://clermontyping.iame-research.center/, accessed on 7 December 2022) [28]. FASTA files were used to determine MLST [29], O and H serotypes [30], and the *fimH* gene [31]. The default parameters were used for each application. For some isolates, the O serotyping was problematic (truncated O genes, missing O genes, or multiple O genes). The O17/O44/O77 problem was resolved according to Joensen et al. [30]; if the wzy variant is O17/O44 and the wzx variant is O17/O77, O17 is the expressed O antigen. For O9/O9a, SeroTypeFinder cannot distinguish these variants [30] and it was reported as O9/O9a. Other problems were investigated with a standard agglutination method [32] for six O serogroups (O8, O9, O46, O92, O101, and O108) to evaluate the somatic phenotype. FASTQ files were used to determine cgMLST [33]. CSIphylogeny [34] on the CGE platform was used to perform multiple alignments with the default parameters with FASTA files. The genome of *E. coli* strain K-12 MG1655 (GenBank acc. num. NC_000913) was used as the reference genome. The SNP phylogenetic tree was annotated with the relevant metadata using iTOL (http://itol.embl.de, accessed on 12 January 2023) [35].

### 2.4. Clonal Lineages and Clones

Only isolates from collection B were considered in the definition of clonal lineages and clones. Isolates with the same phylogroup, serotype, *fimH*, and sequence type (ST) were considered to belong to the same clonal lineage as previously described [23]. The difference in SNPs between each pair of isolates was considered for definition of a clone. 

In the list containing numbers of SNP differences between each pair of isolates (Appendix A), there were a number of SNPs that doubled by a gap in the 12 to 25 SNPs’ differences range values. Therefore, for the setting presented in this study, two isolates separated by 12 SNPs or fewer were considered as clones. The geographically distribution of farms for clonal lineages and clones was mapped in ArcGIS (version 10.8.1). Jittering was used on farm locations to protect their confidentiality, with each farm point being randomly moved within a 5 km radius around their original location.

## 3. Results

### 3.1. Phenotypic Resistance and the Related AMR Genes or AMR-Associated Mutations 

The first hypothesis was that multidrug- or broad-spectrum β-lactam- or fluoroquinolone-resistant *E. coli* isolated from dairy cattle manure possess resistance genes that encode phenotypic resistance. A convenient selection of the most resistant isolates was grouped in collection A to test this hypothesis.

#### 3.1.1. Description of AMR Phenotypes

The most frequent resistance observed according to the breakpoints was to ampicillin (98%), ceftriaxone (90%), sulfisoxazole (88%), ceftiofur (84%), and tetracycline (80%). No resistance was observed towards meropenem. The least frequently observed resistance was to quinolones (ciprofloxacin 15%, danofloxacin 23%, enrofloxacin 22%, and nalidixic acid 6%), azithromycin (17%), and gentamicin (20%) (Table 1).

#### 3.1.2. Description of AMR Genotypes

The predominant AMR genes were *sul*2 (72%), *strA/strB* (65%), *tet(A)* (53%), and *aph(3’)-1a* (47%) (Table 2). Resistance to β-lactams (ampicillin, amoxicillin–clavulanic acid, cefoxitin, ceftriaxone, and ceftiofur) was explained by the presence of nine resistance genes or mutations in low proportions varying between 1 and 43% (Table 2). Of all isolates, 42% carried variants of the *bla_CTX-M_* gene, 41% carried variants of the *bla_CMY-2_* gene, 12% carried a resistance-associated mutation in the *ampC* promoter (-42C > T), and 7% had none of the genes or mutations mentioned above (Table 2). The most frequently observed *bla_CTX-M_* variant in our study was *bla_CTX-M-55_* (15%), followed by *bla_CTX-M-15_* (12%) and *bla_CTX-M-1_* (9%) (Table 2). Three isolates with the *bla_CMY-2_* gene demonstrated a truncated gene between two contigs in the FASTA files (data not shown). For the quinolone class of antimicrobials, *qnrS*1 was the most frequently observed resistance gene (19%). Mutations in *gyrA*, *parC*, or *parE*, associated with decreased susceptibility to fluoroquinolones, were rarely observed (<5%) (Table 2).

#### 3.1.3. Comparison between AMR Phenotypes and Genotypes

The average agreement between the observed AMR phenotypes (defined by the breakpoints in Table 1) and genotypes (defined by the presence of resistance genes or mutations for a specific antimicrobial according to Table 2) was found to be 95%. The average sensitivity (isolates with a resistance gene in the population of isolates with phenotypic resistance) was excellent (97%) and varied between 85% and 100% depending on the antimicrobial tested (Table 3). On a few occasions (*n* = 34), no resistance gene could explain the phenotypic resistance observed. Most of these discrepancies were for β-lactam resistance (82%) (Table 3). It was observed that most of these isolates had intermediate susceptibility results to β-lactam (Table 4). The average specificity (isolates with an absence of a resistance gene in the population of isolates with phenotypic susceptibility) was slightly lower (93%) and also varied according to the antimicrobial tested (52% to 100%). The lowest specificity was observed for streptomycin (52%) and spectinomycin (77%). All phenotype-susceptible genotype-resistance mismatches for these antimicrobials were due to the presence of the resistant genes *aadA*1, *aadA*5, or *strA* + *strB* (Table 4). The majority (57%) of the 76 mismatches with phenotype-susceptible genotype resistance were associated with quinolones. For these antimicrobials, isolates with a resistance gene or resistance-associated mutation showed a small increase in MICs that did not reach clinical breakpoints (Table 4). 

We also used the epidemiological cut-off values (Appendix A) from EUCAST to compare the specificity and sensitivity results obtained. In this way, we wanted to verify whether the use of EUCAST epidemiological cut-off values could improve the lower specificity observed with the breakpoints. Regardless of the values used, there was almost no difference in sensitivity, specificity, or the overall agreement (Appendix A). In an attempt to optimize the overall agreement, we manually tried to find the best fitting MIC values for the presence of the resistance genes for each isolate. The best fitting MIC values (Appendix A) were associated with a sensitivity of 99%, a specificity of 97%, and an overall agreement of 98%.

#### 3.1.4. Comparison between MIC and the Presence of AMR Genes or AMR-Associated Mutations 

For the third-generation cephalosporins (ceftiofur and ceftriaxone), the presence of two resistance genes (*bla_CMY-2_* including *bla_CMY-44_* (*bla_CMY-2_*-like) and *bla_CTX-M_* variants) and a mutation in the *ampC* promoter were associated with phenotypic resistance. There was a slight increase in MIC when a mutation in the *ampC* promoter was observed. However, these MICs were not high enough to reach clinical breakpoints for most isolates (Figure 1). For cephalosporins, the presence of the *bla_CMY-2_* resistance gene was associated with a moderate increase in the MIC whereas the presence of all *bla_CTX-M_* variants was associated with a large increase in the MIC (Figure 1). Epidemiological cut-offs from EUCAST were found to be more accurate than clinical breakpoints in detecting the gene or mutation associated with resistance to third-generation cephalosporins. For fluoroquinolones, multiple resistance-associated mutations in the *gyrA* (p.D87N, p.S83A, p.S83L, and p.S83V), *parC* (p.A56T and p.S80I), and *parE* (p.I355T and p.S458A) genes were observed, as well as the resistance genes *qnrB*19 and *qnrS*1. For three fluoroquinolone-resistant isolates, a combination of multiple resistance genes or resistance-associated mutations was associated with the highest MIC for this class of antimicrobials (Figure 1), indicating a process of accumulation of these genetic modifications associated with a gradual rise in AMR. A high increase in MIC to nalidixic acid was observed when resistance-associated mutations in the *gyrA* gene (p.S83L and p.S83V) were present. However, these mutations only led to a moderate increase in MIC towards fluoroquinolones. It was observed that a mutation in only *parE* (p.I355T) was not associated with an increase in the MIC towards quinolones (Figure 1). The actual clinical breakpoints and EUCAST epidemiological cut-off values were too high to accurately predict any one or a few genes or mutations associated with the resistance to fluoroquinolones.

### 3.2. Mobile Genetic Elements and Proximity of AMR Genes

The second hypothesis was that resistance genes are located close to each other and in the vicinity of mobile genetic elements on the genome. In our study, it is noteworthy that some genes were frequently observed in the same contig at a fixed distance in bp from each other and close to mobile genetic elements (plasmid replicons and insertion sequences) (Figure 2). In this respect, the proximity of the gene *sul*2 located at exactly 61 bp from the genes *aph(6)-1d/aph(3″)-1b* (*strA/strB*) was observed in 30 isolates. The longest fixed distance was observed between the gene *bla_CMY-2_* and a block of five resistance genes located at a distance of exactly 28,341 bp (Figure 2). This proximity was noted in six isolates. An additional observation was that a third-generation cephalosporin resistance gene (*bla_CTX-M-15_*), a resistance gene conferring reduced susceptibility to fluoroquinolones (*qnrS*1), and an insertion sequence (ISKpn19) were in the vicinity of each other. This observation was noted for half of the isolates harboring the *bla_CTX-M-15_* gene (7/14). Taking into consideration all the isolates analyzed (*n* = 118), a total of 365 plasmid replicons were found. Of these isolates, only one did not contain any plasmid replicons. For several isolates, some plasmid replicons were located on the same contigs carrying genes for resistance to antimicrobials (Figure 2). In six isolates, the plasmid replicon Incl1 was located near the resistance gene *bla_CTX-M-1_* conferring resistance to third-generation cephalosporins and the plasmid sequence type of these isolates was the same (pST3). The Incl1 plasmid replicon was also found on the same contig as the *bla_CMY-2_* and ISEc9 genes in three isolates and their pST was identical (pST12). A total of 18 isolates harbored the IncA/C2 (pST3) plasmid replicon but none were on the same contigs as an AMR gene.

### 3.3. Clonal Dissemination between Farms

The third hypothesis of clonal dissemination between farms was tested using a standardized random selection of 86 isolates (representing 86 farms) regrouped in collection B.

#### 3.3.1. Identification of Clonal Lineages, Clones, and Their Associated Characteristics

The reference genome size of *E. coli* K12 is 4,641,652 bp and a total of 3,406,861 bp (73.4%) were identified in all of the analyzed genomes (*n* = 86). A total of 148,561 SNPs were used to construct a phylogenetic tree (Figure 3). A total of 14 clonal lineages (I-XIV) were identified. Clonal lineage I (Phylogroup A, ST10, O101:H9, *fimH*54), consisting of six isolates, was observed on well-dispersed farms in our sample area (Figure 4). The second most widespread clonal lineage, lineage IX (Phylogroup C, ST88, O8:H17, fimH39), consisting of five isolates, was also observed to be well dispersed in the sampling area but more commonly located in the southern half of the map (Figure 4). For these two clonal lineages, different AMR genes were observed in each isolate (data not shown). A total of 10 pairs or triplets of isolates with the same cgMLST were found. All of them had no more than 71 SNPs of difference. Isolates with the same cgMLST had the same ST, serotype, and *fimH*, but were not necessarily clones. Based on our definition of a clone, the maximum number of different SNPs between clonal isolates was set at 12. Three pairs and one triplet of isolates were found to be clones for a total of nine isolates. Interestingly, clone C was observed on two different farms 158 km apart and clone D was observed on three different farms up to 142 km apart (Figure 4). In addition, the two farms on which clone A was detected were geographically close to each other (25 km apart), as well as the two farms on which clone B was detected (6 km apart; Figure 4). Isolates from clones B and D carried the *bla_CTX-M-15_* gene as well as the *qnrS*1 gene, conferring resistance to third-generation cephalosporins and decreased susceptibility to fluoroquinolones, respectively. These two genes were close to each other and in the vicinity of the insertion sequence ISKpn19 (Figure 2).

#### 3.3.2. Phylogroup and β-Lactam Resistance

In our study, it was noted that some phylogroups were associated with specific mechanisms of resistance to β-lactams, in particular to third-generation cephalosporins. Indeed, for phylogroups E and F, only variants of the *bla_CTX-M_* gene were found. In phylogroup D, all isolates possessed the *bla_CMY-2_* gene. Regarding the mutation in the *ampC* promoter n. 42C->T, it was only found in phylogroup C (Figure 3).

## 4. Discussion

The main aim of this work was to elucidate the genetic components of the resistance to broad-spectrum β-lactams and fluoroquinolones and to determine how these genes spread among the *E. coli* population in dairy farms. In order to achieve this, three hypotheses were tested. 

The first hypothesis regarding the agreement between phenotypic and genotypic AMR was validated in this study and was consistent with what has been previously described for *E. coli* despite the different bioinformatics tools used in the various studies [18,36]. The lowest level of sensitivity was towards cefoxitin, and this does not appear to have been described previously. In contrast, a sensitivity of 100% was reported [18] for this antimicrobial, although intermediate isolates were included in the susceptible category. On the other hand, in our study, an intermediate isolate was considered resistant to this antimicrobial. If we consider intermediate isolates as susceptible to cefoxitin the sensitivity increases from 85% to 97%. For third-generation cephalosporins, the problem with sensitivity was found to be associated with the mutation in the *ampC* n.-42C>T promoter. This mutation in *ampC* has previously been associated with a slight increase in the MIC for some third-generation cephalosporins (cefotaxime and ceftazidime) and a strong increase in the MIC for other cephalosporins (ceftazidime and cefoxitin) [37]. In our study, this mutation was associated with a slight increase in the MIC for the third-generation cephalosporins tested (ceftiofur and ceftriaxone) and was not described as being “genotypic resistance” according to ResFinder [25]. If the mutation in the *ampC* n.-42C>T promoter is considered as genotypic resistance, the sensitivity increases to 100% at the cost of a strong decrease in specificity to 42% and 67% for ceftiofur and ceftriaxone, respectively. 

The lowest specificity was observed towards streptomycin. Such genotypically resistant but phenotypically susceptible isolates have also been described previously [18]. The breakpoint seemed to be responsible for this discordance. There is no CLSI breakpoint for this antimicrobial and the CIPARS [11]-NARMS [10] breakpoint (R > 32 μg/mL) and the epidemiological cut-off value (R > 16 μg/mL) seemed to be too high to adequately distinguish the genotype–phenotype agreement for this antimicrobial. In one study, a cut-off value for streptomycin of ≤8 μg/mL for susceptibility (R > 8 μg/mL) was proposed for *E. coli* [38]. In our study, this value was also the best fitting value for the agreement between the phenotype and the genotype for this antimicrobial. If we used this value for streptomycin, its specificity increased from 52% to 93% and its sensitivity remained at 100%. 

Genes and mutations associated with fluoroquinolone resistance also demonstrated low specificity. This was expected because the presence of a single mutation (*gyrA* or *parC*) or resistance gene (*qnr*) was used for the definition of a resistant isolate in our study. It is well known that, for fluoroquinolones, an accumulation of mutations and resistance genes is required to increase the MIC to reach clinical breakpoints [39]. In our study, an accumulation of four resistance genes and/or mutations were observed in the isolates (*n* = 3) associated with the highest MICs to fluoroquinolones. The epidemiological cut-off values yielded better specificity for this class of antimicrobials. A lower breakpoint for fluoroquinolones could be used to determine a better agreement between phenotypic and genotypic resistance when a single mutation or resistance gene is identified. It was observed that most of our isolates harbored a single mutation and were associated with only a slight increase in MIC toward fluoroquinolones. These isolates are not detected by most surveillance programs [10,11] because the clinical breakpoint is used and several mutations or genes are needed to reach this breakpoint. Lower thresholds would therefore be necessary to identify and monitor this type of genotypic resistance to fluoroquinolones.

One limitation of the present study was the definition of phenotypic AMR described by breakpoints. This method of comparison between breakpoints (phenotypic AMR) and resistance genes (genotypic AMR) has been used in numerous studies [16,17,18]. In the present study, the phenotypic AMR was considered as the “gold standard” but this assumption is not exactly true. The breakpoints used are mainly extrapolated from human or cattle respiratory pathogens. This is not necessarily representative of *Enterobacterales* from the gastrointestinal tract of cattle. However, there is no clear method to predict the clinical outcome when administering a specific antimicrobial for bacteria in a specific animal. Phenotypic techniques are tools to predict clinical outcomes but are not perfect. Sequencing techniques could add a plus value to breakpoints currently used for bacteria in veterinary medicine.

The second hypothesis concerning the proximity between resistance genes and mobile genetic elements was also validated in a number of isolates, for some resistance genes and mobile genetic elements. The close proximity between the AMR genes *bla_CMY-2_*, *sul*2, *strA/strB*, *floR*, and *tet(A)* observed in a few of our isolates (*n* = 6) has previously been reported in pig farms from the province of Québec [8]. This latter study also demonstrated, using transformants, that the IncA/C plasmid was associated with phenotypic resistance to β-lactams (*bla_CMY-2_*), sulfisoxazole, streptomycin, chloramphenicol, and tetracycline [8]. However, it was not possible for us to associate the IncA/C plasmid replicon with these resistance genes. This observation highlights the limitation of our study associated with the short-read sequencing technique we used, which does not allow the full description of the plasmids. Indeed, the prediction of large plasmids containing repeated sequences is difficult with short-read sequencing [40]. It has previously been reported that the precision of PlasmidFinder is very high (1.0), but the recall (defined as the percentage of the reference plasmid(s) covered by the prediction) was very low (0.36) [40]. In other words, PlasmidFinder does not detect many plasmids, but false positives are uncommon. In our study, we found multiple plasmid replicons and some of them were on a same contig and close to some AMR genes suggesting that these genes were most likely found on a plasmid. For example, the plasmid replicon Incl1 was located near the resistance gene *bla_CTX-M-1_*. It was also observed that *bla_CTX-M_* variants were in the vicinity of other important AMR genes, such as *qnr*, a gene responsible for an increased MIC towards fluoroquinolones. The *bla_CTX-M-15_* gene was found on the same contig and at exactly 4641 pb from the *qnrS* gene. The number of genes near each other was probably underestimated due to the sequencing method using short-read sequences. Repeated sequences are, indeed, difficult to reconstruct with short-read sequences. 

The mutation in *ampC* promoter n. 42C->T was only found in isolates of the phylogroup C. Because mutations are vertically transferred, this was expected in closely related isolates. In contrast, the resistance gene *bla_CMY-2_* was found in many different phylogroups. This gene is commonly found on plasmids [41,42] which are transferred horizontally. In our study, *bla_CMY-2_* was frequently associated with the same plasmid replicon. However, many plasmid STs were found, suggesting that several different plasmids can harbor *bla_CMY-2_* in our *E. coli* population. The genes *bla_CMY-2_* and *bla_CTX-M_* are associated with resistance to third-generation cephalosporins, which are key antimicrobials of very high importance used to treat humans and animals [43]. In our study, we found a high proportion of these two resistance genes. However, it is difficult to compare our results with those of other studies because a selective protocol was used to collect a proportion of our isolates. In 2007, only *bla_CMY-2_* was reported in animals from Canada and there were no reports of *bla_CTX-M_* at that time [44]. The first report of *bla_CTX-M_* in animals in North America (dairy cattle in Ohio) was published in 2010 [45]. Subsequently, many reports have indicated the presence of *bla_CTX-M_* in animals and animal products [3,4,5,6]. 

The most common *bla_CTX-M_* variant found in this study was *bla_CTX-M-55_*. This is consistent with a large study on *bla_CTX-M_* variants in Canada in which the variant *bla_CTX-M-55_* was found in several animal species [46]. Furthermore, *bla_CTX-M-55_* was the most commonly found variant in beef cattle in Canada [46]. Our results seem to indicate that this situation is also true for dairy cattle in Québec. We also observed close proximity, for some isolates, between the resistance genes *bla_CTX-M-1_* or *bla_CMY-2_* and plasmid Incl1 pST3 or pST12, respectively. This observation was previously reported in pathogenic *E. coli* in pigs in the province of Québec where the resistance genes *bla_CTX-M_* and *bla_CMY-2_* were found on isolates with plasmids Incl1, IncF1B, and IncF1C [6]. A European study on raw chicken meat also reported the presence of these types of *E. coli* (*bla_CTX-M-1_* on plasmid IncI1-Iγ pST3 and *bla_CMY-2_* on plasmid IncI1-Iγ pST12) [47]. These resistance genes are of concern for public health as they could potentially spread to other bacteria, including human and veterinary pathogens [48]. Indeed, it has been suggested that *bla_CMY-2_* could be transmitted between animals and humans via IncI1 and IncK plasmids [49].

Prior to 2019, third-generation cephalosporins, such as ceftiofur, were used frequently on Québec dairy farms, whereas fluoroquinolones were used only sporadically [19]. The resistance to fluoroquinolones could possibly persist in the *E. coli* population via the co-selection process between the *bla_CTX-M-15_* and *qnrS* genes and be mobile due to the proximity of ISkpn19. The proximity of these two genes has been previously observed in *Salmonella* [50]. Close proximity between *qnrS*1, *bla_CTX-M-15_*, *tet(A),* and *sul*2 was also observed in our study, suggesting that the use of tetracyclines or sulfonamides could select for resistance to fluoroquinolones and third-generation cephalosporins. In February 2019, a new regulation was put in place in the province of Québec to better manage the use in production animals of antimicrobials of very high importance in human medicine according to Health Canada [43]. Following this regulation, it was observed that use of category I antimicrobials, such as fluoroquinolones and third-generation cephalosporins, had drastically dropped in dairy cattle [51]. However, no reduction in resistance to these antimicrobials was observed in the same population sampled two years after the implementation [52]. Considering the phenomenon of co-selection, a reduction in resistance to these antimicrobials could be slow and potentially unlikely, given its possible maintenance through the use of other antimicrobials. 

The third hypothesis regarding a clonal transmission of those resistant bacteria between dairy farms was also confirmed. It could be argued that our definition of clones was arbitrary. However, there is not a clear definition of a clone in the literature in regard to the difference of SNPs between isolates. This number varies according to bioinformatics tools used and from one study to another. We used FASTA files for the phylogenic analyses and there was no SNP pruning with this method [34]; therefore, the number of SNPs was overestimated. It is also difficult to compare between studies; using a previously described formula [23], we found that a 10 SNP difference could be used for clonal definition. In other studies, a difference of 10 SNPs [53] or 17-25 SNPs [54] was used to define clonality. However, these numbers were not suitable for this study but gave us an order of magnitude. We considered 12 SNPs in our clonal definition to be conservative. Some clones were found on farms that were geographically far apart from each other, suggesting a dispersal of resistant clones between dairy farms. It is tempting to speculate that the transport of live animals between farms could have spread these clones. However, some farms in our study did not report purchasing or transporting live animals. Transmission by indirect contact would also be possible. It has also been shown that migratory birds can be carriers of *E. coli bla_CTX-M_* [55]. This is consistent with the results of our study, which shows that two of our isolates are ST117 (*fimH*97 and H4), a sequence type associated with poultry [56]. Another study also suggested clonal spread via free-range birds [57]. It has previously been described that calves carry more resistant *E. coli* than adult dairy cows [20]. This age difference could help explain, in part, the spread of clonal resistant bacteria. The exact explanation for this dissemination of clones resistant to broad-spectrum β-lactam and fluoroquinolone antimicrobials remains uncertain. More studies are warranted to understand the dissemination mechanism and the appropriate fighting strategies to tackle this phenomenon.

## 5. Conclusions

In conclusion, this study characterized a population of *E. coli* associated with a high level of multidrug resistance to antimicrobials or resistance to broad-spectrum β-lactams or fluoroquinolones from dairy cattle manure. This study found a strong agreement between AMR phenotypes and genotypes. This work also found that an important proportion of AMR genes could be disseminated vertically and possibly horizontally and that co-selection could explain the persistence of certain resistance profiles. Clones were found on farms geographically far apart from each other and were resistant to both broad-spectrum β-lactam and fluoroquinolone antimicrobials. More studies are warranted to understand the dissemination mechanism, including long-read sequencing to clarify the structure of mobile genetic elements, the appropriate fighting strategies, and the impact of this phenomenon with time. 

## Figures and Tables

**Figure 1 vetsci-10-00242-f001:**
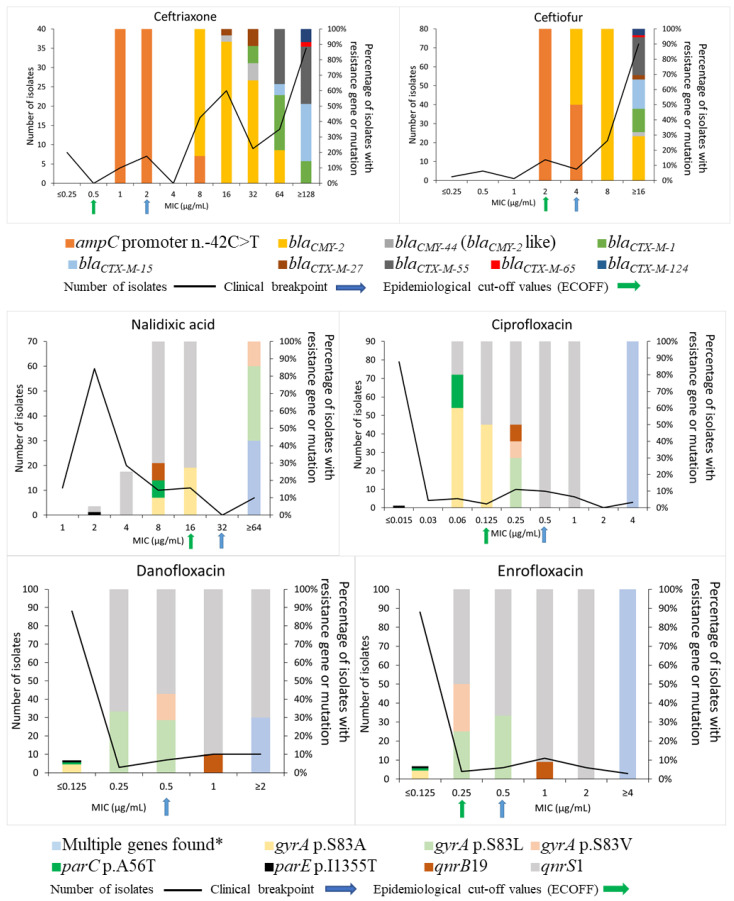
Correlation between an increase in the minimum inhibition concentration (MIC) and the presence of an antimicrobial resistance gene or resistance-associated mutation for third-generation cephalosporins and (fluoro)quinolones for a selection of the most resistant *Escherichia coli* (*n* = 118) isolated from manure at 101 dairy farms in Québec, Canada. * Three isolates harboured multiple mutation points with or without an antimicrobial resistance gene: isolate 10090027-CTX (*gyrA* p.D87N, *gyrA* p.S83L, *parC* p.S80I, and *parC* p.A56T), isolate 10640027-CTX (*gyrA* p.D87N, *gyrA* p.S83L, *parC* p.S80I, and *parE* p.S458A), and isolate 10740013-CTX (*gyrA* p.D87N, *gyrA* p.S83L, *parC* p.S80I, and *qnrS*1).

**Figure 2 vetsci-10-00242-f002:**
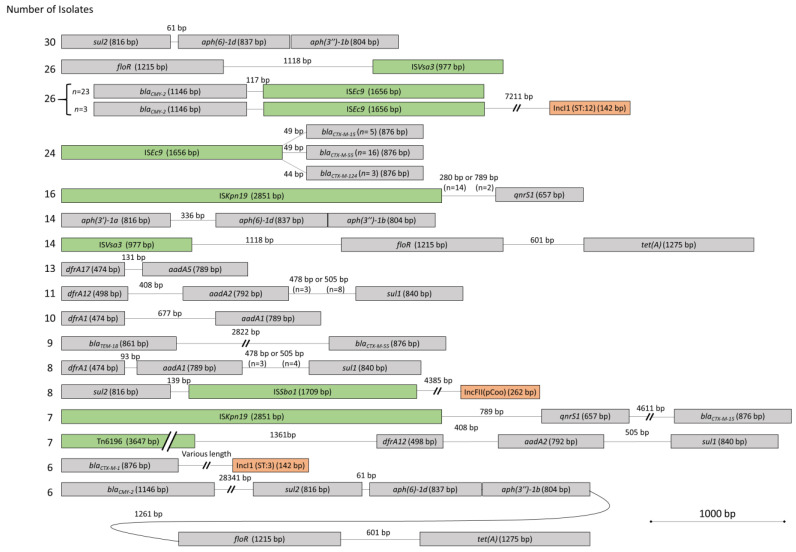
Fixed contiguous resistance genes and mobile genetic elements found with ResFinder, PointFinder, PlasmidFinder, and MGE finder from the Center for Genomic Epidemiology platform for a selection of the most resistant *Escherichia coli* (*n* = 118) isolated from manure from 101 dairy farms in Québec, Canada. Only isolates of which six or more harbour the resistance gene and mobile genetic element are represented in the figure.

**Figure 3 vetsci-10-00242-f003:**
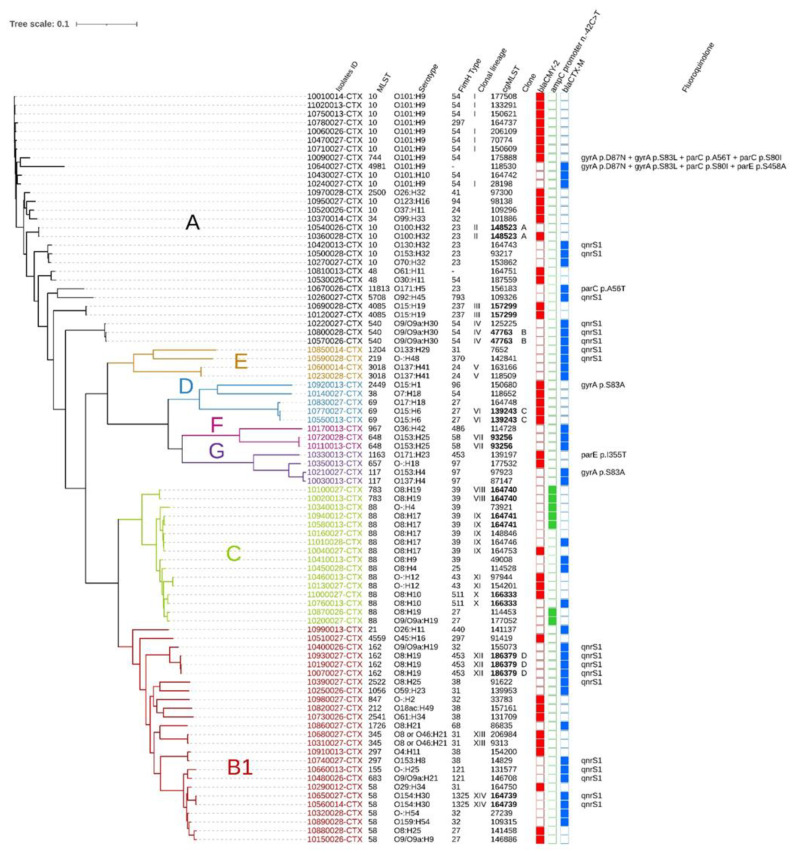
Phylogenetic tree created by CSIPhylogeny from the Center for Genomic Epidemiology by *E. coli* (*n* = 86). Single nucleotide polymorphisms (SNPs) were analysed and modified with the Itol online platform. Each branch with a bootstrap value <1 was deleted. The letters and their corresponding colour to the left are the Clermont phylogroup mash group created by the clermontyping online platform. Isolate ID: the first four digits of the isolate ID number are the farm identification (example: 10890013 = farm #89) and the last four digits are the sample type and the sample period (0012 = cow spring 2017, 0013 = calf spring 2017, 0014 = manure pit spring 2017, 0026 = cow fall 2017, 0027 = calf fall 2017, and 0028 = manure pit fall 2017). The definition of clonal isolates was 12 or fewer SNP differences from each other. A clonal lineage comprises isolates with the same sequence type (ST), serogroup, phylogroup, and *fim*H gene.

**Figure 4 vetsci-10-00242-f004:**
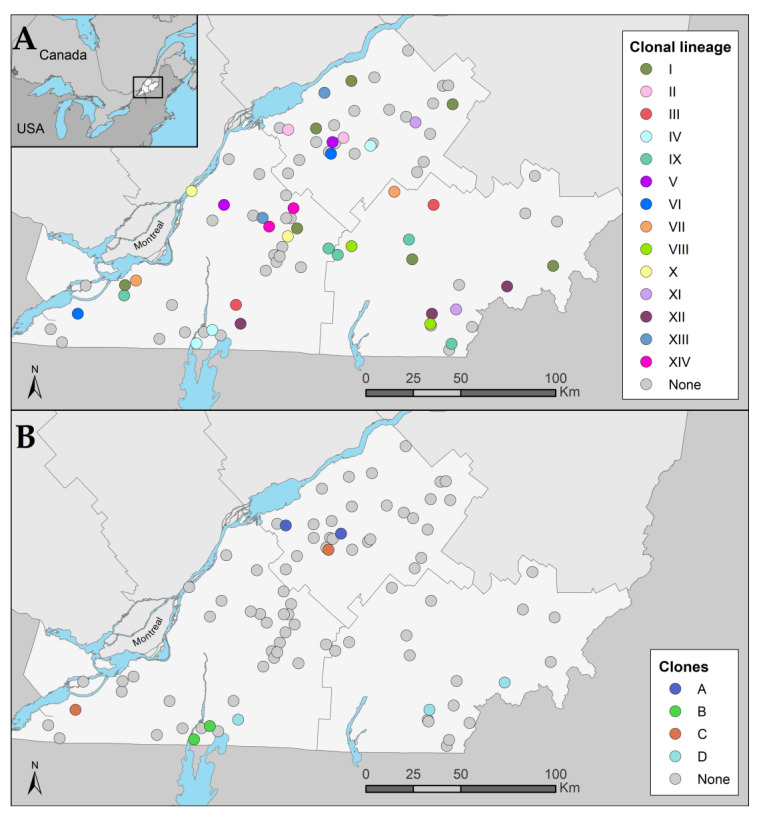
Geographical distribution of isolates according to the definition of: (**A**) Clonal lineage: isolates with the same sequence type (ST), serogroup, phylogroup, and fimH gene. (**B**) Clone: isolates with 12 or fewer SNP differences from each other. Jittering was used for the farm location to preserve their confidentiality. The light grey zone represents the three administrative regions of the study area: Montérégie, Centre-du-Québec, and Estrie. The medium grey zone represents the other administrative regions of Quebec. A Lambert conformal conic projection (NAD 1983) was used for mapping.

**Table 1 vetsci-10-00242-t001:** Minimum inhibitory concentration for a selection of the most resistant *Escherichia coli* (*n* = 118) isolated from manure from 101 dairy farms in Québec, Canada.

Antimicrobial Class	Antimicrobial Agent	MIC (µg/mL)																Non-Susceptible (%)
		0.015	0.03	0.06	0.12	0.25	0.5	1	2	4	8	16	32	64	128	256	512	
Aminoglycoside/aminocyclitol	Gentamicin					2.5	54.2	22.0	0.8			0.8	19.5					20.3
	Neomycin									50.0	0.8	4.2	9.3	35.6				49.2
	Spectinomycin										2.5	28.0	6.8	10.2	52.5			62.7
	Streptomycin									6.8	5.1	5.9	3.4	6.8	72.0			78.8
β-lactam	Amoxicillin/clavu.								1.7	1.7	40.7	3.4	50.0	2.5				55.9
	Ampicillin								1.7					98.3				98.3
	Cefoxitin								4.2	23.7	11.0	9.3	20.3	31.4				61.0
	Ceftiofur					1.7	4.2	0.8	9.3	5.1	17.8	61.0						83.9
	Ceftriaxone					6.8		3.4	5.9		14.4	20.3	7.6	11.9	29.7			89.8
	Meropenem			100.0														0.0
Folate pathway antagonist	Sulfisoxazole											8.5	2.5	0.8			88.1	88.1
	Trimethoprim/sulfa.				16.1	5.1	1.7	0.8	0.8		75.4							75.4
Macrolide	Azithromycin							0.8	12.7	51.7	12.7	5.1	12.7	4.2				16.9
Phenicol	Chloramphenicol									16.9	33.1	1.7		48.3				50.0
	Florfenicol^a^								3.4	33.9	17.8	44.9						NA
Quinolone	Ciprofloxacin	66.9	3.4	4.2	1.7	8.5	7.6	5.1		2.5								15.3
	Danofloxacin				74.6	2.5	5.9	8.5	8.5									22.9
	Enrofloxacin				74.6	3.4	5.1	9.3	5.1	2.5								22.0
	Nalidixic acid							9.3	50.0	16.9	8.5	9.3		5.9				5.9
Tetracycline	Tetracycline									20.3			2.5	77.1				79.7

The numbers represent the percentage of isolates in each MIC category by the antimicrobial agent. White areas represent the concentrations of antimicrobials tested by the broth microdilution method. Percentages in grey areas have an MIC superior to the concentration range tested. Percentages in the first white area starting from the left have an MIC inferior or equal to the corresponding concentration. Dashed and plain lines represent thresholds used to define intermediate and resistant breakpoints. Non-susceptible % is the addition of intermediate and resistant isolates. Yellow areas represent the lowest concentration for non-wild-type isolates according to the European epidemiological cut-off values from EUCAST; ^a^ Florfenicol has no valid clinical breakpoints.

**Table 2 vetsci-10-00242-t002:** Proportion positive for AMR genes or AMR-associated mutations with their related phenotypic resistance among a selection of the most resistant *Escherichia coli* (*n* = 118) isolated from manure from 101 dairy farms in Québec, Canada.

AMR Gene or Mutation	Proportion (%)	AMR Phenotype ^a^	Antimicrobial Class
*aac(3)-IIa*	2	GEN	Aminoglycosides /Aminocyclitols
*aac(3)-IId*	17	GEN	
*aac(3)-Via*	2	GEN	
*aadA*(1,2,5,12,22,24)	27,25,11,1,13,1 ^b^	STR, SPT	
*aph(3′)-Ia*	47	NEO	
*aph(3″)-Ib (strA)*	65	STR	
*aph(6)-Id (strB)*	65	STR	
*ampC* promoter n.-42C>T	12	AMC, AMP, FOX	β-lactams
*bla_CARB-2_*	1	AMP	
*bla_CMY-2_*	39	AMC, AMP, FOX, TIO ^c^, CRO ^d^	
*bla_CMY-44_* (*bla_CMY-2_* like)	2	AMC, AMP, FOX, TIO ^c^, CRO ^d^	
*bla_CTX-M-(1,15,27,55,65,124)_*	9,12,2,15,1,3 ^b^	AMP, TIO ^c^, CRO	
*bla_OXA-1_*	1	AMC, AMP	
*bla_OXA-10_*	1	AMP	
*bla_TEM-1A_*	2	AMP	
*bla_TEM-1B_*	43	AMP	
*dfrA*(1,5,7,8,12,14,16,17,23)	16,8,4,1,19,20,1,11,5 ^b^	SXT	Folate pathway antagonist
*sul*(1,2,3)	38,72,18 ^b^	FIS	
*mph(A)*	21	AZM	Macrolide
*catA*1	9	CHL	Phenicol
*cmlA*1	4	CHL	
*floR*	43	CHL, FFC	
*gyrA* (p.D87N, p.S83A, p.S83L, p.S83V)	3,3,5,1 ^b^	CIP, DAN ^c^, ENR ^c^, NAL	Quinolone
*parC* (p.A56T, p.S80I)	2,3 ^b^	CIP, DAN ^c^, ENR ^c^, NAL	
*parE* (p.I355T, p.S458A)	1,1 ^b^	CIP, DAN ^c^, ENR ^c^, NAL	
*qnrB*19	1	CIP, DAN ^c^, ENR ^c^	
*qnrS*1	19	CIP, DAN ^c^, ENR ^c^	
*ARR*-2	13	Not tested	Rifampin
*tet(A)*	53	TET	Tetracycline
*tet(B)*	36	TET	

^a^ Antimicrobial resistance phenotype according to ResFinder [25] associated with each gene or mutation; ^b^ multiple proportions in a row are associated with each variant of the corresponding AMR gene or mutation in a respective order; ^c^ TIO, DAN, and ENR are not listed in the antimicrobial list of ResFinder; therefore, CRO was assigned for TIO and CIP was assigned for DAN and ENR; ^d^ CRO is not reported to be resistant when a *bla_CMY_* is found but it was considered resistant according to Tyson et al. (2017) [18]; AMC, amoxicillin–clavulanic acid; AMP, ampicillin; AZM, azithromycin, CHL, chloramphenicol; CIP, ciprofloxacin; CRO, ceftriaxone; DAN, danofloxacin; ENR, enrofloxacin; FFC, florfenicol; FIS, sulfisozaxole; FOX, cefoxitin; GEN, gentamicin; MEM, meropenem; NAL, nalidixic acid; NEO, neomycin; SPT, spectinomycin; STR, streptomycin; SXT, trimethoprim–sulfamethoxazole; TET, tetracycline; TIO, ceftiofur.

**Table 3 vetsci-10-00242-t003:** Agreement (number of isolates) observed between antimicrobial resistance phenotypes and genotypes for a selection of the most resistant *Escherichia coli* (*n* = 118) isolated from manure at 101 dairy farms in Québec, Canada.

	Phenotype: Susceptible		Phenotype: Resistant		Sensitivity (%)	Specificity (%)
Antimicrobial	Genotype Resistant	Genotype Susceptible	Genotype Resistant	Genotype Susceptible		
Gentamicin	0	94	24	0	100	100
Neomycin	0	60	56	2	97	100
Spectinomycin	10	34	74	0	100	77
Streptomycin	12	13	93	0	100	52
Amoxicillin/clavu.	0	52	62	4	94	100
Ampicillin	0	2	116	0	100	100
Cefoxitin	0	46	61	11	85	100
Ceftiofur	0	19	96	3	97	100
Ceftriaxone	0	12	96	10	91	100
Meropenem	0	118	0	0	NA	100
Sulfisoxazole	0	14	104	0	100	100
Trimethoprim/sulfa.	4	25	89	0	100	86
Azithromycin	6	92	19	1	95	94
Chloramphenicol	1	58	57	2	97	98
Nalidixic acid	6	105	7	0	100	95
Ciprofloxacin	18	82	18	0	100	82
Danofloxacin	9	82	27	0	100	90
Enrofloxacin	10	82	26	0	100	89
Tetracycline	0	24	93	1	99	100

**Table 4 vetsci-10-00242-t004:** List of discordance observed between antimicrobial resistance phenotypes and genotypes.

Antimicrobial(s)	Pheno.	Geno.	Gene(s)	nb	Explanation
NEO	R	S		2	Unknown
AMC	R	S		4	Unknown, only four isolates with intermediate susceptibility
FOX	R	S		11	Unknown, most have intermediate susceptibility
CRO, TIO	R	S	*ampC ^a^* (-42 C->T)	10	Gene not reported to be associated with a phenotypic resistance to CRO or TIO
AZM	R	S		1	Unknown
CHL	R	S		2	Unknown, only two isolates with intermediate susceptibility
TET	R	S		1	Unknown
SPT	S	R	*aadA*1 or *aadA*5	10	Increase in MIC but not enough to reach clinical breakpoint
STR	S	R	*aadA(*1,5) or *strA* /*strB*	12	Increase in MIC but not enough to reach clinical breakpoint
SXT	S	R	*dfrA*	4	Increase in MIC but not enough to reach clinical breakpoint
AZM	S	R	*mph(A)*	6	Increase in MIC but not enough to reach clinical breakpoint
CHL	S	R	*catA1*	1	Unknown, lowest coverage among *catA*1 (possible non-functional gene)
NAL, CIP, DAN, ENR	S	R	*qnr, gyrA ^a^, parC ^a^* or *parE ^a^*	18	Increase in MIC but not enough to reach clinical breakpoint or silent mutation

^a^ Mutations in these genes are associated with antimicrobial resistance, AMC, amoxicillin–clavulanic acid; AZM, azithromycin, CHL, chloramphenicol; CIP, ciprofloxacin; CRO, ceftriaxone; DAN, danofloxacin; ENR, enrofloxacin; FOX, cefoxitin; NAL, nalidixic acid; NEO, neomycin; SPT, spectinomycin; STR, streptomycin; SXT, trimethoprim–sulfamethoxazole; TET, tetracycline; TIO, ceftiofur.

## Data Availability

The sequences are publicly available from the NCBI and through the bioproject ID PRJNA716674.

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
