# Peer review of "Description of Antimicrobial-Resistant Escherichia coli and Their Dissemination Mechanisms on Dairy Farms"

_vetsci, 2023, doi:10.3390/vetsci10040242_

Round 1
Reviewer 1 Report
This study highlights the dissemination of resistant E. coli clones among dairy farms in Quebec, Canada, and the potential transmission of resistance genes to broad-spectrum β-lactam and fluoroquinolone antibiotics. The authors have conducted a detailed characterization analysis that should be appealing to readers. The entire discussion and experimental design are well done. Here are two main suggestions:
1. Although there are two separate suggestions, they both refer to the same issue. The author presents a lot of discussion about drug resistance genes and mobile genetic elements in the second hypothesis. However, as the author points out, the limitation of this study is that the second-generation short-read sequencing is not suitable for this task. It is suggested that a few strains that need to clarify the sequence structure be selected to complete the second and third generation sequencing to achieve whole-genome assembly. At present, there is quite a lot of literature showing that this is not difficult to accomplish. 2. The third hypothesis raises many concerns about the authors' conclusion. The authors claim that the detection of a cephalosporin resistance gene, a fluoroquinolone resistance gene, and an insertion sequence in the same clonal isolates from distant farms suggests a possible role of mobile genetic elements in the spread of AMR in dairy cattle. These strains clearly need to undergo whole-genome assembly to answer the author's hypothesis.Author Response
Thank you for your kind evaluation regarding our manuscript. Your suggestion to include long read sequencing is relevant. Unfortunately, this was not included in the budget for this project and this technique can be time consuming and very expensive. This project has now been completed and all the available funds have been used for the WGS presented in the present manuscript. However, we agree that this is an excellent suggestion for a future project. To address your concern about the absence of long-read sequencing in the conclusion, the last sentence of the abstract has been modified :
“Furthermore, these clones are capable of disseminating resistantce to broad-spectrum β-lactam and fluoroquinolone antimicrobials.
The conclusion was also modified :
“This work also found that an important proportion of AMR genes could be disseminated vertically and possibly horizontally, and that co-selection could explain the persistence of certain resistance profiles. Clones were found on farms geographically far apart from each other and could disseminate resistance were resistantce to both broad-spectrum β-lactam and fluoroquinolone antimicrobials. More studies are warranted to understand the dissemination mechanism, including long-read sequencing to clarify the structure of mobile genetic elements, the appropriate fighting strategies and the impact of this phenomenon with time.”
With these modifications, the conclusion is consistent with the results obtained with short-read sequencing.
Hoping to answer your concern,
Reviewer 2 Report
Dear authors
From line 94 to 97 pass it to the end of the discussion or in the conclusion
Divide the paragraph of line 123-151, 254 -279, 435-460, in general all paragraphs with more than 10 lines
There are very similar sections between the methodology of the manuscript and the article entitled "Prevalence of Antimicrobial Resistance and Characteristics of Escherichia coli Isolates From Fecal and Manure Pit Samples on Dairy Farms in the Province of Québec, Canada", review and only refer to the article published in 2021, to avoid having equal sections.
Write the objective and conclusion in third person
Author Response
Thank you for your good evaluation. All your comments have been integrated in the manuscript:
For the first comment on the last lines of the introduction (line 94-97) (this comment was also noted by reviewer 4). This section was added according to instruction for authors of Veterinary sciences "Introduction: .... Finally, briefly mention the main aim of the work and highlight the main conclusions..." However, we understand your point and we simply remove those lines.
Long paragraphs were also divided. For the paragraph at line 123-151 of the methodology, this paragraph was shortened to respond to your next comment about a similar section of methodology. Paragraph of line 254-279, 435-460, and 509-539 were divided.
The methodology has been properly shortened/ cited and reformulated to avoid having equal sections.
Objectives, hypotheses and conclusion are now in the third person (this work, this study, etc…)
Reviewer 3 Report
The article deals with the elucidation of the genetic components of the resistance to broad-spectrum β-lactams and fluoroquinolones and the determination of how these genes spread among the E. coli population in dairy farms.
This is a very interesting study that undoubtedly shed light on the problem and makes important contributions in this field.
It is also a very honest manuscript that clearly recognizes and exposes its limitations.
I have however some issues that should be solved before final acceptance:
CONCRETE POINTS:
-Line 20: Please, delete “or E. coli”. It is redundant.
-Line 162: To be determined? Please, clarify and place the appropriate information.
-Lies 175-176: Please, give more detailed information on the solving of these problems (O serotyping).
-The legends of the foots of the Tables and Figures should be written in letter smaller than the rest of the text in order to clearly differentiate it. In the same way, title of the Table 3 should be properly separated from the previous text.
-Figure 4: Would it be possible to point out on the maps any city or important population center for a better location and understanding, especially for non-Canadian readers?
Author Response
Thank you very much for your great evaluation.
E. coli at line 20 has been deleted.
The Bioproject ID is PRJNA716674 and has been added to the manuscript. Please note that all biosamples (n=118) are associated with this bioproject but sequences are still processing in SRA. All sequences will be publicly available once they are process. This should be in the next few days.
For lines 175-176 (O serotyping), this explanation has been added to the manuscript : The O17/O44/O77 problem was resolved according to Joensen et al. [30]; if the wzy variant is O17/O44 and the wzx variant is O17/O77, O17 is the expressed O antigen. For O9/O9a, SeroTypeFinder cannot distinguish those variants [30] and it was reported as O9/O9a. Other problems were investigated with a standard agglutination method [32] for six O serogroups (O8, O9, O46, O92, O101, and O108) to evaluate the somatic phenotype.
As suggested, the legends of tables and figures are now in smaller letters. Table 3 is now separated from the text.
For figure 4, Montréal has been added to the map as well as a small map of Canada/USA for a better location.
Reviewer 4 Report
The manuscript "Description of antimicrobial resistant Escherichia coli and their dissemination mechanisms on dairy farms" focus an important subject related to antimicrobial resistance mechanisms and their impact in the “One Health” perspective.
The Simple Summary could be resumed.
Line 16 – “The World Health Organization has proclaimed antimicrobial resistance to be one of the greatest current threats to global health.” This sentence is unnecessary.
Lines 21-22 – “This technology predicts 95% of the antimicrobial resistance compared to the classic technique.” This sentence is unnecessary in the simple summary.
The Abstract is correct and elucidates the content of the manuscript.
Line 39 and all manuscript – Change concordance by agreement.
The introduction section is satisfactory.
The main question addressed by the research is clear in lines 86-90 of introduction.
Lines 93-97 – “An understanding of this dissemination mechanism will allow elaboration of fighting strategies to control resistant bacteria. Our study reveals the dissemination of resistant E. coli clones between dairy farms. Furthermore, these clones are capable of disseminating resistance to broad-spectrum β-lactam and fluoroquinolone antimicrobials.” Should be removed from the introduction section.
The materials and methods seem adequate. The study implies vast laboratorial work with different and complex techniques, that must be considered.
The results are sound. The tables and figures may be a little bulky.
The discussion is correct.
Author Response
Thank you for your good evaluation.
As you suggested, the simple summary has been resumed. Sentences of the lines 16 and 21-22 were deleted.
The word concordance was changed to agreement in manuscript (8 modifications).
For the comment on the last lines of the introduction (line 94-97) (this comment was also noted by reviewer 2), this section was added according to instruction for authors of Veterinary sciences "Introduction: .... Finally, briefly mention the main aim of the work and highlight the main conclusions..."
However, we understand your point and we simply remove those lines as you suggested.